# Trends and Application of Artificial Intelligence Technology in Orthodontic Diagnosis and Treatment Planning—A Review

Farraj Albalawi [1,2,*] and Khalid A. Alamoud [1,2]

1   Preventive Dental Science Department, College of Dentistry, King Saud Bin Abdulaziz University for Health Sciences, Riyadh 11426, Saudi Arabia
2   King Abdullah International Medical Research Centre, Ministry of National Guard Health Affairs, Riyadh 11481, Saudi Arabia
*   Correspondence: balawif@ksau-hs.edu.sa

**Abstract:** Artificial intelligence (AI) is a new breakthrough in technological advancements based on the concept of simulating human intelligence. These emerging technologies highly influence the diagnostic process in the field of medical sciences, with enhanced accuracy in diagnosis. This review article intends to report on the trends and application of AI models designed for diagnosis and treatment planning in orthodontics. A data search for the original research articles that were published over the last 22 years (from 1 January 2000 until 31 August 2022) was carried out in the most renowned electronic databases, which mainly included PubMed, Google Scholar, Web of Science, Scopus, and Saudi Digital Library. A total of 56 articles that met the eligibility criteria were included. The research trend shows a rapid increase in articles over the last two years. In total: 17 articles have reported on AI models designed for the automated identification of cephalometric landmarks; 12 articles on the estimation of bone age and maturity using cervical vertebra and hand-wrist radiographs; two articles on palatal shape analysis; seven articles for determining the need for orthodontic tooth extractions; two articles for automated skeletal classification; and 16 articles for the diagnosis and planning of orthognathic surgeries. AI is a significant development that has been successfully implemented in a wide range of image-based applications. These applications can facilitate clinicians in diagnosing, treatment planning, and decision-making. AI applications are beneficial as they are reliable, with enhanced speed, and have the potential to automatically complete the task with an efficiency equivalent to experienced clinicians. These models can prove as an excellent guide for less experienced orthodontists.

**Keywords:** artificial intelligence; automated diagnosis; digital diagnosis; supervised learning; orthodontics; dento-facial orthopedics; deep learning; machine learning; artificial neural networks; convolutional neural networks

## 1. Introduction

Artificial intelligence (AI) is a new breakthrough in technological advancements based on the concept of simulating human intelligence. AI models are designed to assist humans in completing tasks more efficiently and are developed through training and programming, using a large number of datasets, and then executing these models to turn the data into actionable information in performing a specific task [1]. These emerging technologies highly influence the diagnostic process in medical sciences, with enhanced accuracy in diagnosis [2]. AI models have been applied in healthcare in order to assist healthcare professionals and enhance the accuracy in diagnosis, decision making, and predicting prognosis, with the ultimate goal of enhancing patient care and treatment outcomes. AI models have also demonstrated performances that are on par with trained and experienced healthcare professionals [3].

AI technology has been widely applied in dentistry. These models are based on Machine Learning (ML) systems, which are designed using algorithms that can be trained

using a large number of data sets and later applied to new sets of data for testing and evaluation. ML based AI models have displayed excellent accuracies in making predictions based on the datasets without human interventions [4–6]. AI models based on the newly developed and advanced system known as Deep Learning (DL) are designed to mimic the functioning of the neural system of the human brain; hence they are termed artificial neural networks [7].

The AI system has been gaining high popularity in dentistry, as they have been widely applied for diagnosing and predicting oral diseases such as dental caries, periodontal diseases, and oral cancer [8–11]. AI models have demonstrated exceptional performances in diagnosis, determining the treatment needs, and predicting the prognosis of diseases [1,5,8–13]. Along with this, AI models have also been designed and applied in the field of forensic odontology, where the models have demonstrated excellent precision in performing tasks [14]. Considering these developments in AI, this review article intends to report on the trends and application of AI models designed for diagnosis and treatment planning in orthodontics.

## 2. Materials and Methods

### 2.1. Search Strategy

Data search for the articles that have reported on the application of AI models in orthodontics was carried out in the most renowned electronic databases. These databases mainly included PubMed, Google Scholar, Web of Science, Scopus, and Saudi Digital Library. This search was mainly carried out for original research articles published over the last 22 years, from 1 January 2000 until 31 August 2022. Several Medical Subheadings (MeSH terms) were used for searching the articles in the electronic databases. MeSH terms included: artificial intelligence, automated diagnosis, computer-assisted diagnosis, digital diagnosis, supervised learning, orthodontics, dentofacial orthopedics, unsupervised learning, diagnosis, prognosis, prediction, deep learning, machine learning, artificial neural networks, and convolutional neural networks. Boolean operators AND/OR were used to generate a combination of these key words in the advanced search. Language filters for the English language were also used. A manual search for articles was also performed simultaneously in the college central library after screening the references of the articles obtained from the electronic search.

At this stage, article selection was based on the title and abstract; 628 articles related to our research topic were extracted. Later, 436 articles were excluded due to duplication. The remaining 192 articles were applied for the eligibility criteria for being included in this review article.

### 2.2. Eligibility Criteria and Study Selection

Original research articles that reported on the application of AI models in orthodontics were included in this systematic review. Articles with no full text, narrative reviews, scoping reviews, letters to editors, opinion letters, case reports, short communications, conference proceedings, and articles other than English, were excluded (Figure 1).

A total of 56 articles that fulfilled the eligibility criteria were subjected to qualitative synthesis (Table 1).

**Table 1.** Details of the studies that reported in application of AI based models in orthodontics.

| Serial No | Authors and Reference | Year of Publication | Type of Algorithm Architecture | Objective of the Study | No. of Patients/Images/ Photographs for Training Testing | Study Factor | Modality | Comparison If Any | Evaluation Accuracy/ Average Accuracy/ Statistical Significance | Results (+)Effective, (−)Non Effective (N) Neutral | Limitations of the Study | Authors Suggestions/ Conclusions |
|---|---|---|---|---|---|---|---|---|---|---|---|---|
| 1 | Nishimoto s et al. [15] | 2019 | CNNs | Automatic cephalometric landmark detection | 153 samples for training, 66 for validating | Landmarks | Lateral cephalogram | Specialist | Not clear | (N) neutral | CNNs based model predicted cephalometric analysis were not significantly different from those plotted manually | This model still in the state of development |
| 2 | Park JH et al. [16] | 2019 | ANNs | Automatic identification of cephalometric landmarks | 1028 samples for training, 283 for validating | Landmarks | Lateral cephalogram | Benchmarks in the literature | YOLOv3 algorithm outperformed SSD in accuracy | (+)Effective | The accuracy was inferior to other methods when the size of objects is small | This model can be of great use for use in clinical practice. |
| 3 | Chen S et al. [17] | 2020 | ML | Automatic landmarks identification | 60 samples | Landmarks | Cone-Beam Computed Tomography (CBCT) images | Not mentioned | Not clear | (+)Effective | Not mentioned | Fast and efficient CBCT image segmentation will be analyzed more efficiently |
| 4 | Kunz F et al. [18] | 2020 | CNNs | Automated cephalometric X-ray analysis | 50 samples | Landmarks | Cephalometric X-rays | 12 experienced examiners | No statistically significant differences | (+)Effective | Not mentioned | Results were of the same quality level as experienced examiners |
| 5 | Hwang HW et al. [19] | 2020 | CNNs | Automated identification of cephalometric landmarks | 1028 samples for training and 283 samples for testing | Landmarks | Cephalograms | Human examiners | Accuracy similar to human examiners | (+)Effective | When the data set was less than 500 the AI model did not identify the landmarks correctly | Larger quantity of datasets will be required in the future. |
| 6 | Zeng M et al. [20] | 2020 | CNNs | Automatic cephalometric landmark detection | 150 for training dataset and 250 test images | Landmarks | Cephalograms | Not mentioned | Significant performance | (+)Effective | The model needs improvement in the future work. | This is a good model for detecting landmarks |
| 7 | Lee JH et al. [21] | 2020 | BCNNs | Locating cephalometric landmarks | 150 images for training, 250 test images | Landmarks | Cephalograms | Two expert Examiners | Significantly higher performance | (+)Effective | The model was trained on regional geometrical features only | Improves the accuracy and reliability of decisions of the specialists |
| 8 | Bulatova G et al. [22] | 2021 | CNNs | Automatic cephalometric landmark identification | 110 samples | Landmarks | Cephalograms | Senior orthodontic resident | No statistical difference between the two | (+)Effective | The operator is supposed to put a digital ruler which can be subjected to human errors. | Can increase efficiency in routine clinical practice |

**Table 1.** *Cont.*

| Serial No | Authors and Reference | Year of Publication | Type of Algorithm Architecture | Objective of the Study | No. of Patients/Images/ Photographs for Training Testing | Study Factor | Modality | Comparison If Any | Evaluation Accuracy/ Average Accuracy/ Statistical Significance | Results (+)Effective, (−)Non Effective (N) Neutral | Limitations of the Study | Authors Suggestions/ Conclusions |
|---|---|---|---|---|---|---|---|---|---|---|---|---|
| 9 | Hwang HW et al. [23] | 2021 | CNNs | Automated cephalometric analysis | 1983 for training, 200 for testing | Landmarks | Cephalograms | Expert Examiner | Superior than Human examiner | (+)Effective | The performance of the model can be affected with the noise issues inherent in medical imaging. | Superior performance than those reported in literature |
| 10 | Kim J et al. [24] | 2021 | CNNs | Automated identification of cephalometric landmarks | 3150 for training, 100 for testing | Landmarks | Cephalograms | Two orthodontists | Significant performance | (+)Effective | Only hard tissue landmarks for training | This model can replace human task |
| 11 | Kim YH et al. [25] | 2021 | CNNs | Automatic cephalometric landmark identification | 800 for training, 100 for testing | Landmarks | Cephalograms | Two Calibrated examiners | Significant performance | (+)Effective | Not mentioned | This model achieved better results than examiners |
| 12 | Kim MJ et al. [26] | 2021 | CNNs | Automatic cephalometric landmark identification | 860 samples | Landmarks | CBCT images | One experienced orthodontist | Significant performance | (+)Effective | Did not compare the prediction accuracy of a model trained by a more experienced clinician | This model showed better consistency than manual identification |
| 13 | Kim MJ et al. [27] | 2021 | CNNs | Automatic cephalometric landmark identification | 860 samples 80% training, 20% validating | Landmarks | CBCT images | One experienced orthodontist | Significant performance | (+)Effective | Amount of data required to achieve the expected accuracy could not be explained | This model showed superior results compared to previous studies |
| 14 | Yao J et al. [28] | 2022 | CNNs | Automatic cephalometric landmark location | 512 samples training, 200 for testing and validating | Landmarks | Cephalograms | Two experienced orthodontists | Higher accuracy | (+)Effective | Amount of data volume was less and need to increased | This model meets the requirements of different cephalometric analysis methods. |
| 15 | Le VNT et al. [29] | 2022 | CNNs | Human–AI collaboration for the identifying cephalometric landmarks | 1193 samples training, 100 for testing | Landmarks | Cephalograms | Twenty dental students | Accuracy was higher than dental students | (+)Effective | Amount of dataset was small and obtained for very young patients | This beginner–AI collaboration model was effective in detecting the landmarks |
| 16 | Gil SM et al. [30] | 2022 | CNNs | Automated identification of cephalometric landmarks | 2075 samples for training, 343 for validating | Landmarks | Cephalograms | One experienced examiner | Demonstrated an high successful detection rate | (+)Effective | The comparison was made with one single examiner | This model is an effective alternative to manual identification |

**Table 1.** *Cont.*

| Serial No | Authors and Reference | Year of Publication | Type of Algorithm Architecture | Objective of the Study | No. of Patients/Images/ Photographs for Training Testing | Study Factor | Modality | Comparison If Any | Evaluation Accuracy/ Average Accuracy/ Statistical Significance | Results (+)Effective, (−)Non Effective (N) Neutral | Limitations of the Study | Authors Suggestions/ Conclusions |
|---|---|---|---|---|---|---|---|---|---|---|---|---|
| 17 | Dot G et al. [31] | 2022 | CNNs | Automatic localization ocephalometric landmarks | 160 samples for training, 38 for validating | Landmarks | Computed tomography (CT) scans | One experienced operator | Excellent agreement with the human examiner | (+)Effective | This model still requires improvement as the data sets were limited | This reliability of the model is par with that of the clinician |
| 18 | Kök H et al. [32] | 2019 | ANNs | Determining growth and development by cervical vertebrae stages | 300 samples | Reference points | Cephalometric radiographs | One trained orthodontist and Seven different AI algorithms | This model displayed second-highest accuracy | (+)Effective | More datasets needed for training and evaluation | This model can be used as decision support to clinicians |
| 19 | Kök H et al. [33] | 2020 | ANNs | Growth and development periods and gender from the cervical vertebrae | 419 samples 70% for training, 15% testing and, 15% for validating | Reference points | Cephalometric and hand-wrist radiographs | Researcher | Displayed high accuracy | (+)Effective | More datasets needed for training and evaluation | The success of this model was satisfactory |
| 20 | Kök H et al. [34] | 2020 | ANNs | Determining growth and development based on cervical vertebra ratios | 360 samples | Reference points | Cephalometric radiographs | Naïve Bayes models (NBMs) | More successful than the reference model | (+)Effective | Datasets belonged to one population, need to study different and multi-racial | This model was more successful than the previous models |
| 21 | Amasya H et al. [35] | 2020 | ANNs | Determining cervical vertebral maturation (CVM) analysis | 647 samples | Reference points | Cephalometric radiographs | One examiner Five different ML models | Best results was achieved by ANN model | (+)Effective | Absence of hand-wrist radiographs | This model can be used for prediction of cervical vertebrae morphology |
| 22 | Amasya H et al. [36] | 2020 | ANNs | Cervical vertebral maturation analysis | 647 samples | Reference points | Cephalometric radiographs | Three experienced dentomaxillofacial radiologists and one experienced orthodontist | Displayed better performance | (+)Effective | The data obtained was from the wide age range (10–30 years) | This model performed close to or even better than human observers |
| 23 | Seo H et al. [37] | 2021 | CNNs | Cervical vertebral maturation analysis | 600 samples | Reference points | Lateral cephalometric radiographs | Experienced radiologist, Six deep learning models | All models demonstrated excellent accuracy Inception-ResNet-v2 performing the best | (+)Effective | Small number of data set used | This model will help practitioners in making accurate diagnoses and treatment plans |

**Table 1.** *Cont.*

| Serial No | Authors and Reference | Year of Publication | Type of Algorithm Architecture | Objective of the Study | No. of Patients/Images/ Photographs for Training Testing | Study Factor | Modality | Comparison If Any | Evaluation Accuracy/ Average Accuracy/ Statistical Significance | Results (+)Effective, (−)Non Effective (N) Neutral | Limitations of the Study | Authors Suggestions/ Conclusions |
|---|---|---|---|---|---|---|---|---|---|---|---|---|
| 24 | Zhou J et al. [38] | 2021 | CNNs | Cervical vertebral maturation status | 1080 samples (980 for training and 100 for testing) | Reference points | Cephalometric radiographs | Two experienced examiners | There was a good agreement between AI the Human examiners | (+)Effective | Smaller size of data set for testing | This model is a useful and reliable tool for assessing CVM. |
| 25 | Kim DW et al. [39] | 2021 | ML | Predicting the hand-wrist maturation stages based on the cervical vertebrae (CV) images, | 499 samples | Reference points | Hand-wrist radiographs and lateral cephalograms | Three specialists | Better prediction accuracy | (+)Effective | Smaller size of data set along with more pediatric patients | This model can aid as a decision supporting tool |
| 26 | Kim EG et al. [40] | 2022 | CNNs | Estimating cervical vertebral maturation | 600 samples | Reference points | Lateral cephalograms | Four experienced specialists | Demonstrated best accuracy | (+)Effective | Datasets were developed using radiographs from single institution | This model displayed best accuracy and is of practical applicability |
| 27 | Mohammad-Rahimi H et al. [41] | 2022 | CNNs | Cervical vertebral maturation (CVM) degree and growth spurts | 890 samples | Reference points | Lateral cephalometric radiographs | Two orthodontists | Substantial agreement between the experienced examiners and the AI model | (+)Effective | Improvements need to be done in data quality | This model can provide practical assistance to practicing dentists |
| 28 | Li H et al. [42] | 2022 | CNNs | Estimating cervical vertebral maturation | 6079 samples (70% for training, 15% testing and, 15% for validating) | Reference points | Cephalometric radiographs | Two experienced orthodontists ResNet152, DenseNet161, GoogLeNet, VGG16 | ResNet152 demonstrated best accuracy | (+)Effective | Quality and quantity of the datasets was severely affected | This model can be used as an automatic auxiliary diagnostic tool |
| 29 | Atici SF et al. [43] | 2022 | CNNs | Classification of the Cervical Vertebrae Maturation | 1018 samples | Reference points | Cephalometric radiographs | Expert Orthodontist, CNN, MobileNetV2, ResNet101, and Xception | This CNN model provide higher accuracy than the models | (+)Effective | Not mentioned | This model can be used as effective tool for analyzing the skeletal maturity stage and timing of the treatment. |
| 30 | Croquet B et al. [44] | 2021 | CNNs | Automated land –marking for palatal shape analysis | 1045 samples (732 for training, 209 for testing and 104 for validating) | Landmarks | Dental casts | Single operator | There was no difference between automatic and manual analysis with promising accuracy and reliability, | (+)Effective | The data was of individuals with dentition till second molars may not reflect the true diversity of individuals of interest to orthodontists | This model can be used for land-marking of digitized dental casts for clinical and research purpose. |

Table 1. *Cont.*

| Serial No | Authors and Reference | Year of Publication | Type of Algorithm Architecture | Objective of the Study | No. of Patients/Images/ Photographs for Training Testing | Study Factor | Modality | Comparison If Any | Evaluation Accuracy/ Average Accuracy/ Statistical Significance | Results (+)Effective, (−)Non Effective (N) Neutral | Limitations of the Study | Authors Suggestions/ Conclusions |
|---|---|---|---|---|---|---|---|---|---|---|---|---|
| 31 | Nauwelaers N et al. [45] | 2021 | CNNs | Palatal and dental shape estimation | 1324 samples | Landmarks | 3D Dental casts | Different models | Singular auto-encoder achieved competitive performance in terms of accuracy, generalization, specificity, and variance | (+)Effective | The model was limited to shapes that underwent an elaborate pre-processing | This model can a useful tool for shape analysis in the future |
| 32 | Xie X et al. [46] | 2010 | ANNs | Determining the need for orthodontic tooth extraction | 200 samples (180 for training, 20 for testing) | Indices | Casts and cephalometrics | Humans | 80% accuracy in determining the need for extraction | (+)Effective | Limited amount of samples | This model can be considered a decision-making tool |
| 33 | Jung SK et al. [47] | 2016 | ANNs | Diagnosis of orthodontic tooth extractions | 156 samples (96 for training, 60 for testing) | Indices | Casts and cephalometrics | Experienced orthodontist | High performance Excellent success rates | (+)Effective | Diagnosis of extractions was confined to nonsurgical procedures | Can be used as an new approach in orthodontics |
| 34 | Li P et al. [48] | 2019 | ANNs | Determining the need of orthodontic tooth extraction | 302 samples | Feature variables | Casts and cephalometrics | Two experienced orthodontists | Excellent performance with 94.0% accuracy | (+)Effective | Limited amount of samples | This model can provide a good guidance for less experienced orthodontists. |
| 35 | Choi HI et al. [49] | 2019 | ML | Determining the need of orthodontic tooth extraction | 316 samples | Datasets | Casts and cephalometrics | One experienced orthodontist | High performance with 91% accuracy | (+)Effective | Exclusion of skeletal asymmetry cases | Can be applied for the diagnosis of cases |
| 36 | Suhail Y et al. [50] | 2020 | ML | Diagnosis of orthodontic tooth extraction | 287 samples | Datasets | Casts and cephalometrics | Five experienced orthodontist | In agreement with the experienced orthodontists | (+)Effective | Limited feature set where the treatment outcomes were confined to only non-surgical orthodontic procedures | Can be considered a decision-making tool in clinical practice |
| 37 | Etemad L et al. [51] | 2021 | ML | Decision on orthodontic tooth extraction | 838 samples | Datasets | Casts and cephalometrics | Previous models | Performance was lesser than the previous models | (+)Effective | Not mentioned | This model lacks generalizability and in order to improve it needs advanced artificial intelligence algorithms |

**Table 1.** *Cont.*

| Serial No | Authors and Reference | Year of Publication | Type of Algorithm Architecture | Objective of the Study | No. of Patients/Images/ Photographs for Training Testing | Study Factor | Modality | Comparison If Any | Evaluation Accuracy/ Average Accuracy/ Statistical Significance | Results (+)Effective, (−)Non Effective (N) Neutral | Limitations of the Study | Authors Suggestions/ Conclusions |
|---|---|---|---|---|---|---|---|---|---|---|---|---|
| 38 | Real AD et al. [52] | 2022 | ML | Determining the need of orthodontic tooth extraction | 214 samples | Datasets | Casts and cephalometrics | Two experienced orthodontists | Demonstrated an accuracy of 93.9% | (+)Effective | Degree of over fitting that may have occurred in the models | This model achieved best performance when model and cephalometric data are combined |
| 39 | Yu HJ et al. [53] | 2020 | CNNs | Automated Skeletal Classification | 5890 samples (70% for training, 15% testing and, 15% for validating) | Datasets | Clinical data and cephalometrics | Five experienced orthodontists | Demonstrated an highest accuracy at 96.40% | (+)Effective | The data were collected from a single organization | This model has a potential for skeletal orthodontic diagnosis |
| 40 | Wang H et al. [54] | 2021 | CNNs | Automated multiclass segmentation of the jaw and teeth | 30 samples | Landmarks | CBCT scans | 4 experienced dentists | Accurate in its performance | (+)Effective | Data of complicated dental status need to be considered | This model can reduce the amount of time and effort spent in clinical settings and increase the efficiency and performance of dentists |
| 41 | C.H Lu et al. [55] | 2009 | ANNs | Image prediction post orthognathic surgery (OGS) | 30 samples | Landmarks | Lateral Cephalogram Facial images | Profile post-surgery profile | Very less prediction errors | (+)Effective | Not mentioned | Can be applied for predicting post-surgical facial profile |
| 42 | H. H Lin et al. [56] | 2018 | CNNs | Assessing facial asymmetry in patients undergone OGS | 100 samples | Landmarks | 3D facial images | Specialist | Predications were statistically significant $p < 0.05$ | (+)Effective | Small sample size was used for developing the model | Human like efficient tool for decision making |
| 43 | R. Patcas et al. [57] | 2019 | CNNs | Assessing post OGS facial attractiveness | 146 samples | Landmarks | Facial photographs | Profile post-surgery profile | Was in comparison with the actual improvement | (+)Effective | Dissimilarities between the subjective patient's view and the computed score could exist | Is an efficient tool for assessing facial attractiveness |
| 44 | P. G. M. Knoops et al. [58] | 2019 | CNNs | Diagnosing of OGS | 4261 samples | Landmarks | Data sets 3D face scans | Not mentioned | 95.5% sensitivity, 95.2% specificity, | (+)Effective | Larger data sets needed for training the models | An efficient tool for diagnosing OGS |
| 45 | R.Stehrer et al. [59] | 2019 | CNNs | Predicting perioperative blood loss | 950 subjects | Comparing with actual blood loss | Data sets | Data on actual blood loss | Statistical significance ($p < 0.001$). | (+)Effective | Data for the model was developed from records from one single clinic | An efficient tool for estimating perioperative blood loss |

**Table 1.** *Cont.*

| Serial No | Authors and Reference | Year of Publication | Type of Algorithm Architecture | Objective of the Study | No. of Patients/Images/ Photographs for Training Testing | Study Factor | Modality | Comparison If Any | Evaluation Accuracy/ Average Accuracy/ Statistical Significance | Results (+)Effective, (−)Non Effective (N) Neutral | Limitations of the Study | Authors Suggestions/ Conclusions |
|---|---|---|---|---|---|---|---|---|---|---|---|---|
| 46 | S.H.Jeong et al. [60] | 2020 | CNNs | Predicting soft tissue profiles that require OGS | 822 samples | Landmarks | Facial photographs | 2 orthodontist, 3 maxillofacial surgeons, 1 maxillofacial radiologist. | An Accuracy of 0.893 | (+)Effective | Certain level of false positives and false negatives cases were revealed by the model | An efficient tool predicting soft tissue profiles |
| 47 | K.S. Lee et al. [61] | 2020 | DCNNs | Differential diagnosis of OGS | 220 samples | Landmarks | Lateral Cephalogram | Four different models | Modified-Alexnet displayed an Accuracy of 0.919 | (+)Effective | Comparison was done with a limited data | Modified-Alexnet displayed the highest level performance |
| 48 | C.Tanikawa et al. [62] | 2020 | ANNs | Predicting facial morphology post OGS | 137 samples | Landmarks | Lateral cephalogram and 3-D facial images | 2 AI models | Excellent success rates | (+)Effective | The model was developed and tested with data from only two clinics. | An efficient tool predicting post OGS facial morphology |
| 49 | D. Xiao et al. [63] | 2021 | CNNs | For planning of OGS | 47 samples | Landmarks | CT Scans Clinical data sets | Sparse representation method | Significant ($p$ <0.05). | (+)Effective | The model trained on simulated pairs of deformed-normal bones and the number was limited | This model outperformed an existing sparse representation method |
| 50 | G. Lin et al. [64] | 2021 | CNNs | Assessing the need for OGS in Unilateral Cleft Lip and Palate patients | 56 samples | Landmarks | Lateral Cephalogram | Boruta method | An excellent accuracy of 87.4%. | (+)Effective | The data used was limited and was from a single center | This model is capable of predicting the need for surgery |
| 51 | H.H.Lin et al. [65] | 2021 | CNNs | Assessing pre and post OGS facial symmetry | 71 samples | Landmarks | CBCT images | 4 orthodontists and 4 plastic surgeons and also with previously reported models | Accuracy of 90%. | (+)Effective | This model was trained with a limited data sets | This model exhibited high performance. |
| 52 | L.J. Lo et al. [66] | 2021 | CNNs | Assessing facial soft tissue symmetry before and after OGS | 158 samples | Landmarks | 3-D facial photographs | Pre and post-operative | Statistically Significant | (+)Effective | Dissimilarities might exist between the patient's subjective view and the machine scoring | The model can aid clinicians in assessing facial symmetry |

Table 1. *Cont.*

| Serial No | Authors and Reference | Year of Publication | Type of Algorithm Architecture | Objective of the Study | No. of Patients/Images/ Photographs for Training Testing | Study Factor | Modality | Comparison If Any | Evaluation Accuracy/ Average Accuracy/ Statistical Significance | Results (+)Effective, (−)Non Effective (N) Neutral | Limitations of the Study | Authors Suggestions/ Conclusions |
|---|---|---|---|---|---|---|---|---|---|---|---|---|
| 53 | R.Horst et al. [67] | 2021 | CNNs | Predicting the virtual soft tissue profile post-surgery | 133 samples (119 for training, 14 for testing) | Landmarks | 3D photographs and CBCT images | Mass Tensor Model (MTM) | Statistically significant ($p = 0.02$) | (+)Effective | In asymmetric cases and extreme cranial or caudal displacements, the model under predicted these displacements | This model can accurately predict the soft tissue profile post-surgery |
| 54 | W.S.Shin et al. [68] | 2021 | CNNs | Predicting the need for OGS | 413 samples | Landmarks | Cephalogram | 2 orthodontists, 3 maxillofacial surgeons, 1 maxillofacial Radiologist. | An excellent accuracy of 0.954 | (+)Effective | This model involved only Korean patients from only one hospital | Displayed higher accuracy in predicting the need for OGS |
| 55 | Y.H Kim et al. [69] | 2021 | CNNs | Diagnosing orthodontic surgery | 960 samples (810 for training, 150 for testing) | Landmarks | Cephalogram | ResNet-18, 34, 50, and 101 | Success rate was displayed by ResNet-18 = 93.80%, ResNet-34 = 93.60% | (+)Effective | The data used was from a single center | This model can diagnose whether to conduct orthognathic surgery |
| 56 | G. Dot et. al. [70] | 2022 | CNNs | Multi-task segmentation of cranio-maxillofacial structures for OGS | 453 samples (300 for training, 153 for testing) | Landmarks | CT Scans | 2 Operators | Excellent performance | (+)Effective | Cannot assess the reliability of the results as the data was from one single center | This model need to be trained from other databases for better reliability |

Footnotes: ML = Machine Learning, ANNs = Artificial Neural Networks, CNNs = Convolutional Neural Networks, DCNNs = Deep Neural Networks, Bayesian Convolutional Neural Networks (BCNN), CT- scans-Computed Tomography, CBCT- Cone-Beam Computed Tomography, OCT-Optical Coherence Tomography.

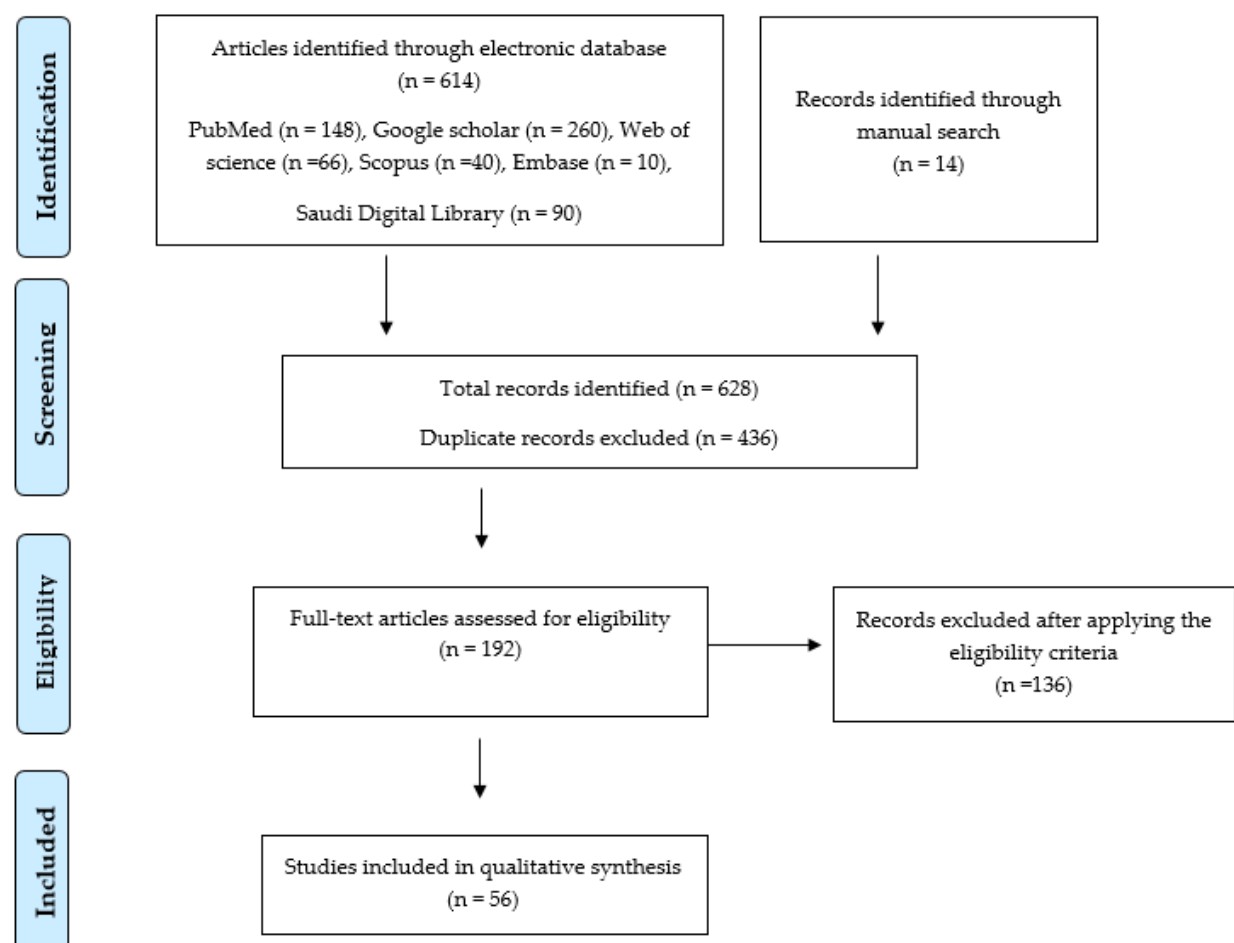

**Figure 1.** Flow chart for screening and selection of articles.

Summary of AI models designed for different diagnostic tasks is presented in Table 2.

**Table 2.** Summary of AI models designed for different diagnostic tasks.

| AI Technique/Algorithm Architecture | Diagnostic Tasks | Functionality of the AI Model | Input Features |
|---|---|---|---|
| Machine Learning (ML) | Automated identification of landmarks | Landmarks | Cone-Beam Computed Tomography (CBCT) images [17] |
| | Predicting the hand-wrist maturation stages based on the cervical vertebrae (CV) images | Reference points | Hand-wrist radiographs and lateral cephalograms [39] |
| | Determining the need of orthodontic tooth extraction | Datasets | Casts and cephalometrics [49–52] |
| Artificial Neural Network (ANNs) | Automated identification of landmarks | Landmarks, Reference points | Lateral cephalogram [16], Cephalograms [18,35] |
| | Cervical vertebral maturation analysis | Reference points | Cephalometric radiographs [36] |
| | Determining growth and development by cervical vertebrae stages | Indices | Cephalometric radiographs [32], Cephalometric and hand-wrist radiographs [33,34] |
| | Determining the need for orthodontic tooth extraction | Indices | Casts and cephalometrics [46–48] |
| | Predicting facial morphology post OGS | Landmarks | Lateral cephalogram and 3-D facial images [62] |

**Table 2.** *Cont.*

| AI Technique/Algorithm Architecture | Diagnostic Tasks | Functionality of the AI Model | Input Features |
|---|---|---|---|
| Deep Learning/ Convolutional Neural Networks (CNNs) | Automated identification of landmarks | Landmarks | Cone-Beam Computed Tomography (CBCT) images [15,18,26,27,31], Cephalograms [19–25,28–30] |
| | Cervical vertebral maturation analysis | Reference points | Lateral cephalometric radiographs [37,38,40], Cephalometric radiographs [42] |
| | Cervical vertebral maturation (CVM) degree and growth spurts | Reference points | Lateral cephalometric radiographs [41] |
| | Classification of the Cervical Vertebrae Maturation | Reference points | Cephalometric radiographs [43] |
| | Automated land –marking for palatal shape analysis | Landmarks | Dental casts [44,45] |
| | Automated Skeletal Classification | Datasets | Clinical data and cephalometrics [53] |
| | Automated multiclass segmentation of the jaw and teeth | Landmarks | CBCT scans [54,70] |
| | Image prediction post orthognathic surgery (OGS) | Landmarks | Lateral Cephalogram Facial images [55] |
| | Assessing facial asymmetry in patients undergone OGS | Landmarks | 3D facial images [56] |
| | Assessing post OGS facial attractiveness | Landmarks | Facial photographs [57] |
| | Diagnosing of OGS | Landmarks | Data sets 3D face scans [58], Lateral Cephalogram [61,68,69], CT Scans and Clinical data sets [63] |
| | Predicting perioperative blood loss | Data sets | Data on actual blood loss [59] |

## 3. Results

### 3.1. Qualitative Synthesis of the Included Studies

The research trend shows a gradual increase in the number of research publications over the last two decades. However, in the last two years, the number of articles reported on the application of AI models in orthodontics has rapidly increased (Figure 2).

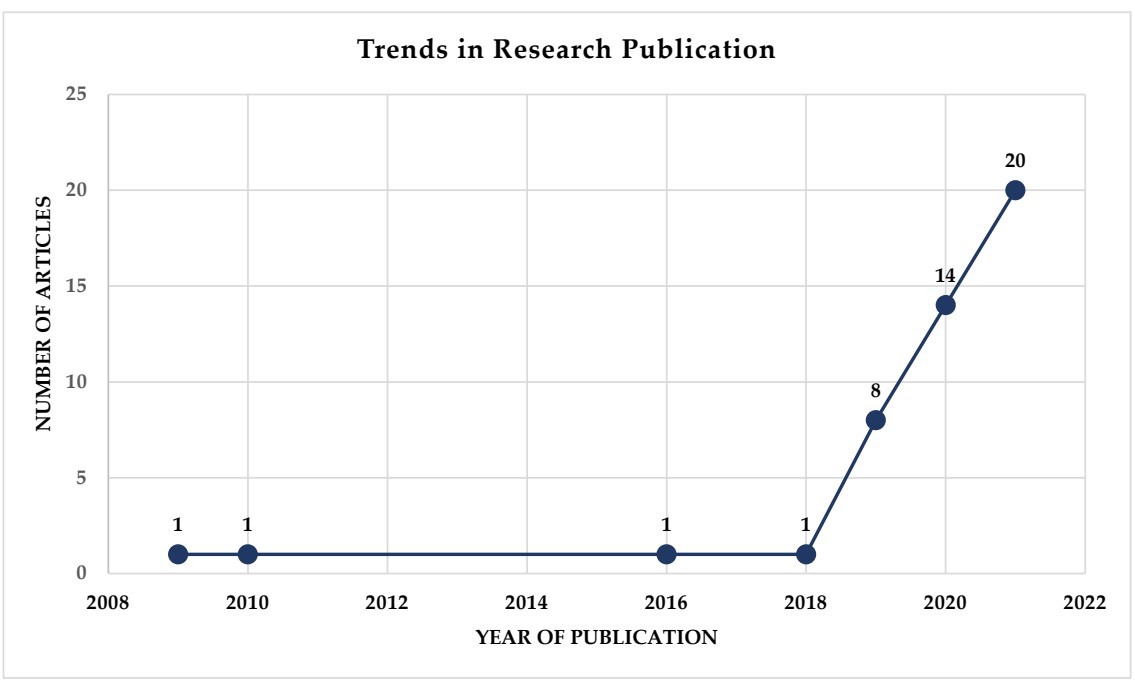

**Figure 2.** Trends in research publications.

*3.2. Study Characteristics*

AI models developed for application in orthodontics have mainly focused on: the automated identification of cephalometric landmarks [15–31]; the estimation of bone age and maturity using cervical vertebra and hand-wrist radiographs [32–43]; palatal shape analysis [44,45]; determining the need for orthodontic tooth extractions [46–52]; automated skeletal classification [53,54]; and the diagnosis and planning of orthognathic surgeries [55–70] (Figure 3).

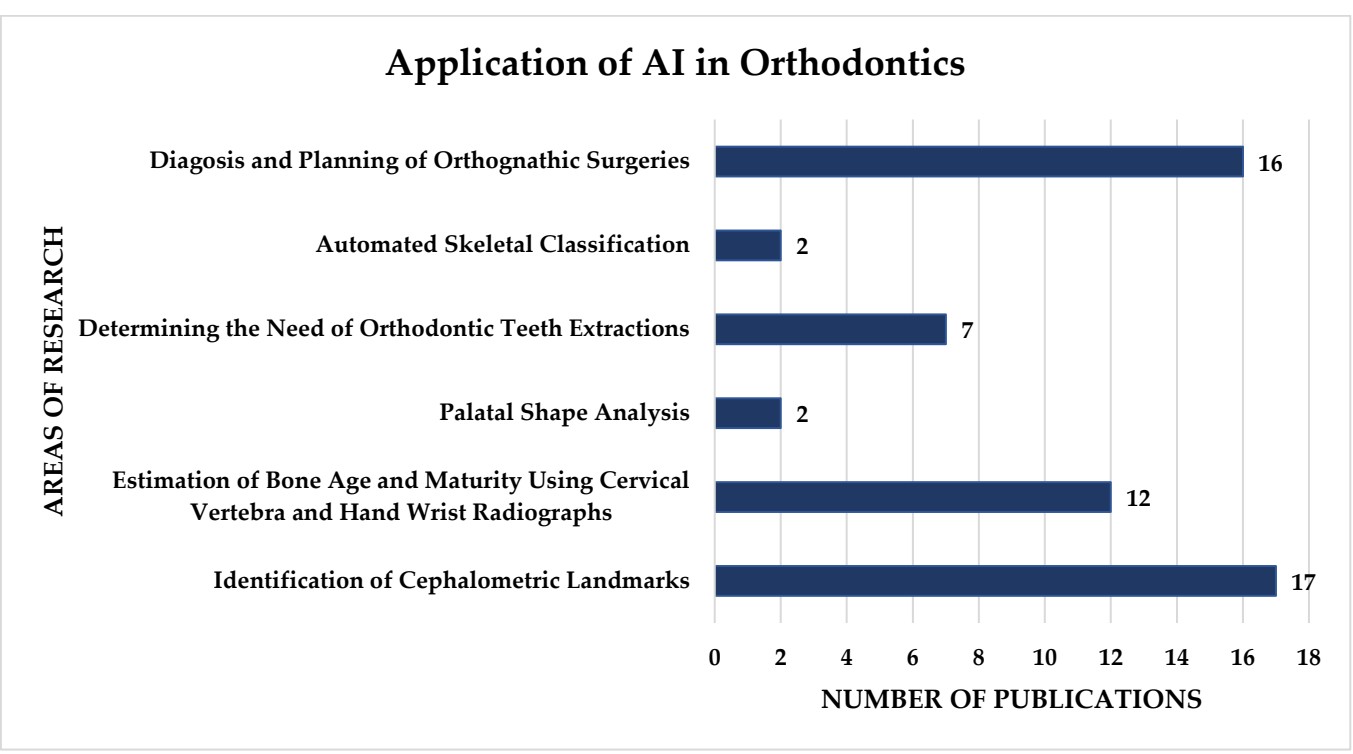

**Figure 3.** Application of AI in orthodontics.

## 4. Discussion

Digitalization in medical diagnosis is a new breakthrough in the field of health sciences. In dentistry, digital technology has been widely applied in clinical practice, especially with the use of 3D intra oral scanners designed for scanning dental arches. These technical advancements have simplified the process of impression making and fabrication of prostheses [71,72]. The recent advancements in the field of medical sciences and computer technology have resulted in significant developments in AI based models that have been designed for diagnosing oral diseases, treatment planning, clinical decision making, and predicting treatment outcomes; these models have demonstrated excellent performances [1,9–11].

*4.1. AI Models Designed for Automatically Identifying Cephalometric Landmarks*

In orthodontics, cephalometric analysis is considered one of the most significant tools for evaluating the skeletal profile of the craniofacial region. The manual tracing of X-ray films and plotting the landmarks requires skill and expertise. Advancements in AI technology have resulted in the development of automated models that can predict landmarks without human assistance. Nishimoto S et al. [15] reported on an automated landmark predicting system based on a deep learning neural network, where the model demonstrated a performance equivalent to manual plotting. Park JH et al. [16] reported on the performance of two automatic deep learning models You-Only-Look-Once version 3 (YOLOv3) and Single Shot Multibox Detector (SSD) designed for identifying cephalometric landmarks. The YOLOv3 algorithm outperformed SSD in accuracy and displayed 5% higher accuracy than

the benchmark methods reported in the literature. Kunz F et al. [18] reported on an automatic cephalometric identification model using an (AI) algorithm; this model demonstrated quality levels equivalent to experienced professional examiners. Hwang HW et al. [19] reported on the performance of an automated identification system, You-Only-Look-Once version 3 (YOLOv3), for the automatic identification of cephalometric landmarks. This model displaced excellent accuracies, which were similar to experienced human examiners. Zeng M et al. [20] reported on an AI based model designed for automatically predicting cephalometric landmarks; this model demonstrated competitive performance compared with other models. Bulatova G et al. [22] reported on an AI based model for automated cephalometric landmark identification; this model demonstrated a performance similar to a calibrated senior orthodontist. Hwang HW et al. [23] compared the performance of a deep learning based AI model with previously reported AI models in the literature designed to identify cephalometric landmarks automatically. This model demonstrated superior results in comparison with the previously reported AI models. Kim J et al. [24] reported that a CNN model designed to identify cephalometric landmarks displayed excellent accuracies. Kim YH et al. [25] reported on a deep learning based fully automatic AI model for identifying cephalometric landmarks; this model demonstrated better performance than two calibrated and experienced examiners. Kim MJ et al. [26] reported on a CNN model designed for the automated identification of cephalometric landmarks using cone-beam computed tomography (CBCT) scans; this model displayed better consistency in identifying the landmarks in comparison with the experienced human examiners. Another study conducted by Kim MJ et al. [27] reported on an automatic cephalometric landmark identification system using CBCT images; this model displayed a performance equivalent to the experienced examiners. Yao J et al. [28] reported on a CNN based AI based model for automatically identifying the cephalometric landmarks; this model demonstrated a higher accuracy. Le VNT et al. [29] reported on a human-AI collaboration for identifying cephalometric landmarks. The collaborative system was effective in identifying landmarks. Gil SM et al. [30] reported on a convolution neural network (CNN) based AI model for identifying cephalometric landmarks; this model demonstrated excellent performance, similar to a human examiner's performance.

### 4.2. AI Models Designed for Bone Age and Maturity Estimation

The timing of orthodontic treatment is crucial in achieving the desired clinical outcomes. In treatment planning, quantifying the skeletal growth, mainly the mandible, impacts the diagnosis, treatment planning, and treatment outcomes [73]. Therefore, if orthodontic treatment is initiated during the optimum development phase, it will produce more favorable results. Otherwise, a much longer treatment time and surgical intervention may be needed to correct the deformities of the jaw [74,75]. The standard method for estimating bone maturity uses a hand-wrist radiograph and a lateral cephalogram to estimate cervical vertebral maturity (CVM). However, studies have reported that the reproducibility of the CVM varies among examiners [76,77]. In recent developments, AI has been widely used to estimate bone maturation using hand-wrist radiographs or CVM. Kok H et al. [32] reported on the efficiency of seven AI classifiers designed to determine growth; based on cervical vertebrae stages. These models demonstrated acceptable performance in determining the stages. Another study conducted by Kok H et al. [33] also reported the application of an artificial neural network (ANN) model for determining growth based on cervical vertebrae. This model demonstrated satisfactory results in determining the growth-development periods. Kok H et al. [34] also reported on the success rates of two AI models, artificial neural network models (NNMs) and naïve Bayes models (NBMs), designed for determining the growth and development based on the cervical vertebrae. NNMs displayed a higher success rate than the NBMs. Amasya H et al. [35] reported on the performance of five AI based Machine Learning (ML) models designed for CVM analysis. These AI models displayed more accuracy in comparison with the human observer. Amasya H et al. [36] also reported on the validation of the ANN model designed for CVM

analysis; this model displayed better performance in comparison with four trained human observers. Seo H et al. [37] reported on the performance of six CNN based deep learning models designed for CVM analysis on cephalometric radiographs. These six models demonstrated more than 90% accuracy in performing the task. Zhou J et al. [38] reported on the performance of an AI model designed for automatically determining the CVM status using cephalometric radiographs. This model demonstrated good agreement with the human examiners. Kim DW et al. [39] reported on the AI model designed for predicting the hand-wrist skeletal maturation stages. The model demonstrated excellent accuracy in predicting the skeletal maturation stages. Kim EG et al. [40] reported on an AI based CNN model designed for estimating the CVM using lateral cephalograms. The model displayed 93% accuracy in performing the task. Mohammad-Rahimi H et al. [41] reported on an AI model for determining CVM and growth spurts using lateral cephalograms. The model demonstrated reasonable accuracy in determining the CVM stage and displayed high reliability in estimating the pubertal stage. Li H et al. [42] reported on AI based CNN model for CVM classification. This model demonstrated good accuracy in classifying the CVM. Atici SF et al. [43] reported on an AI based deep learning model designed for estimating the CVM stages; this model displayed higher accuracy in determining the CVM stages.

### 4.3. AI Models Designed for Palatal Shape Analysis

The palate has a complex structure, and its shape varies among individuals. The palate's shape is related to the facial pattern and a wide range of factors such as breathing pattern and occlusion [78–80]. The shape of the palate is of great interest to orthodontists as it is a potential area for evaluation and assessing the outcome of orthodontic procedures like maxillary expansion [81,82].

Palatal measurements usually include palatal surface area, volume, and depth [78]. These measurements often require greater experience and are often subjected to observer errors. The recent technological advancements have resulted in the development of AI based models designed for palatal shape analysis. Croquet B et al. [44] reported on a deep learning model designed for analyzing the palatal shape. This automatic model demonstrated excellent repeatability with promising accuracy. Nauwelaers N et al. [45] reported on the application of an AI based deep learning model for palatal shape analysis, and this model achieved results similar to conventional approaches.

### 4.4. AI Models Designed for Determining the Need for Extractions

Determining the need for tooth extraction and deciding which teeth need to be extracted is a critical decision in orthodontic treatment planning as it is irreversible [83]. The orthodontists' decision regarding extraction is based on their training, clinical experience, and treatment philosophies [84]. AI technology has been applied to designing models which can be used as an axillary tool for deciding on the need for orthodontic extractions. Xie X et al. [46] reported on the AI based ANN model for determining the need for extraction; this model was very efficient and displayed an accuracy of 80% in determining the need for extraction. Jung SK et al. [47] reported on the performance of an AI model designed to diagnose the need for orthodontic extraction. The model was very efficient in diagnosing extraction and non-extraction cases. Li P et al. [48] reported on the multilayer perceptron ANN model for predicting extraction and non-extraction cases. The model demonstrated an excellent accuracy of 94% for predicting the extraction and non-extraction cases. Choi HI et al. [49] reported on an AI based model designed to determine the need for extraction. This model displayed a success rate of 91% for determining extraction decisions. Suhail Y et al. [50] reported on a machine learning model to diagnose the need for extraction. The model demonstrated a performance that was in agreement with the trained examiners. Etemad L et al. [51] reported on a machine learning model for predicting the need for extraction and non-extraction. The model displayed acceptable results; however, the authors suggested a need for improvisation in the algorithms to improve generalizability.

*4.5. AI Models Designed for Planning Orthognathic Surgeries*

Individuals presenting dentofacial deformities, either due to congenital or acquired conditions, may require orthognathic surgeries in order to reposition the jaws into a functional relationship. In recent years, several articles have reported on the application of automated computerized methods designed for analyzing dentofacial deformities and elaborating treatment plans [85]. Patcas R et al. [57] reported on an AI model designed for predicting facial appearance post orthognathic surgery; this model displayed an acceptable performance in predicting facial attractiveness and appearance. Knoops PGM et al. [58] reported on a machine learning model designed for automated diagnosis and treatment planning. This model demonstrated an excellent sensitivity of 95.5% and specificity of 95.2% in diagnosing the patients. Stehrer R et al. [59] reported on an AI model for predicting perioperative blood loss following orthognathic surgery. The model demonstrated an excellent result and efficiently predicted the perioperative blood loss prior to surgery. Jeong SH et al. [60] reported on an AI model designed for judging the soft tissue profiles requiring orthognathic surgery. The model displayed an accuracy of 89.3% in judging the soft tissue profiles requiring surgery. Lee K-S et al. [61] reported on an AI based deep CNN model for differential diagnosis of orthognathic surgery. The model was successful and can be applied for differential diagnosis of orthognathic surgery. Tanikawa C et al. [62] reported on an AI model designed for predicting facial morphology post orthognathic surgery. The model demonstrated an excellent success rate in predicting facial morphology and can be applied for clinical purposes. Xiao D et al. [63] reported on an AI model designed for virtually simulating the surgical plan. The model showed higher accuracy in generating the shape models. Shin W et al. [68] reported on an AI model that automatically predicts the need for orthognathic surgery. The model was efficient with relative accuracy in predicting the need for surgery. Kim YH et al. [69] reported on an AI based deep learning model designed to diagnose orthognathic surgery. The model demonstrated excellent performance in predicting the diagnosis of orthognathic surgery.

A few of the limitations of this paper might be with the search strategy. Even though we have performed a comprehensive search for articles, some might have been missed. In general, these AI models' limitations are mainly due to the limited amount of data sets that have been applied for training these models, and validating and testing. Another limitation of the data sets is the standardization since the data sets applied for assessing the performance of these AI models are obtained from one diagnostic center in most cases. Hence, the performance of these models may vary when exposed to different data sets from multiple centers. However, considering the performance of these AI models, there is an urgent need to develop and implement policies to accelerate the process of approval of these models for marketing and usage in clinical scenarios, which can help clinicians in the diagnosis and decision making process.

## 5. Conclusions

The past decade has witnessed tremendous advancements in digital diagnostic techniques. AI is a major development that has been successfully implemented in a wide range of image-based applications. These applications can facilitate clinicians in diagnosing, treatment planning, and decision making. These applications are extremely useful as they are reliable and fast methods that have the potential of automatically completing the task with an efficiency equivalent to experienced clinicians. These models can prove to be an excellent guide for less experienced orthodontists. However, there are a few limitations with most of these models, with respect to the limited number of datasets used for training and validating these models, and the reliability of the data, as they are obtained from a single hospital/institution or a single machine. Hence, greater improvisation needs to be conducted in this area for better reliability and generalizability.

**Author Contributions:** Conceptualization, F.A.; methodology, F.A. and K.A.A.; software, F.A.; validation, F.A.; formal analysis, F.A.; investigation, F.A.; resources, F.A.; data curation, F.A.; writing—original draft preparation, F.A. and K.A.A.; writing—review and editing, F.A. and K.A.A.; visualization, F.A.; supervision, F.A.; project administration, F.A. All authors have read and agreed to the published version of the manuscript.

**Funding:** This research received no external funding.

**Institutional Review Board Statement:** Not applicable.

**Informed Consent Statement:** Not applicable.

**Data Availability Statement:** Not applicable.

**Conflicts of Interest:** The authors declare no conflict of interest.

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
