# Peer review of "Trends and Application of Artificial Intelligence Technology in Orthodontic Diagnosis and Treatment Planning—A Review"

_applsci, doi:10.3390/app122211864_

Round 1

Reviewer 1 Report

This article reviews the trends and application of artificial intelligence technology in Orthodontic diagnosis and treatment planning. Some minor comments.

It is better to add one section discussing overall and limitations before the conclusion with the latest literature.

In the Discussion. Please discuss regarding the intraoral scanners used for the scanning of the dental arch.

https://pubmed.ncbi.nlm.nih.gov/34552983/

https://file.techscience.com/uploads/attached/file/20201029/20201029063712_40569.pdf

Author Response

Dear Reviewer,
Greetings!

Firstly I would like to thank you for your valuable inputs, and would like to inform you that we have considered all the valuable comments suggested by you and have modified the manuscript as per your suggestions.

We are also providing point to point clarifications for the comments suggested by you.  "Please see the attachment."

We have modified the manuscript to best of our knowledge, kindly consider the same and oblige,

Thank you and regards

Sincerely yours,

Dr. Farraj Albalawi (Corresponding Author)

Reviewer 1

It is better to add one section discussing overall and limitations before the conclusion with the latest literature.

Response: Thank you for your valuable suggestion, we have added them as per the reviewer’s suggestion

In the Discussion. Please discuss regarding the intraoral scanners used for the scanning of the dental arch.

https://pubmed.ncbi.nlm.nih.gov/34552983/

https://file.techscience.com/uploads/attached/file/20201029/20201029063712_40569.pdf

Response: Thank you for your valuable suggestion, we have added a paragraph on the intraoral scanners used for the scanning of the dental arch and added the suggested reference article

Reviewer 2 Report

Dear Author

the paper is interesting and it is about a very actual topic.

Some little English typing errors are evident.

Even if this is a review I think that the references are too many, it could be useful to reduce the reviwe on the articles of last ten years.

Author Response

Dear Reviewer,
Greetings!

Firstly I would like to thank you for your valuable inputs, and would like to inform you that we have considered all the valuable comments suggested by you and have modified the manuscript as per your suggestions.

We are also providing point to point clarifications for the comments suggested by you.  "Please see the attachment."

We have modified the manuscript to best of our knowledge, kindly consider the same and oblige,

Thank you and regards

Sincerely yours,

Dr. Farraj Albalawi (Corresponding Author)

Reviewer 2

The paper is interesting and it is about a very actual topic.

Some little English typing errors are evident.

Even if this is a review I think that the references are too many, it could be useful to reduce the review on the articles of last ten years.

Response: Thank you for your valuable suggestions. We have considered all your valuable suggestions and made got the manuscript edited for English language. We have considered these references as they were very informative and valuable.

Reviewer 3 Report

Please reformat the table 1 so it is easier to be read. Preferably create Table 2 to summarized table 1 tabulation.

Besides those Table can conclusion be equipped with summary table (TABLE 3) for each of the functionality of the AI, which methods of AI being used (CNN, ANN etc) and what's the findings.

Author Response

Dear Reviewer,
Greetings of the day!

Firstly I would like to sincerely thank you for your valuable inputs, and would like to inform you that we have considered your valuable comments and have modified the manuscript as per your suggestions.

We are also providing point to point clarifications for the comments suggested by you. "Please see the attachment."

We have modified the manuscript to best of our knowledge, kindly consider the same and oblige,

Thank you and regards

Sincerely yours,

Dr. Farraj Albalawi (Corresponding Author)

Reviewer 3

Please reformat the table 1 so it is easier to be read. Preferably create Table 2 to summarized table 1 tabulation.

Besides those Table can conclusion be equipped with summary table (TABLE 3) for each of the functionality of the AI, which methods of AI being used (CNN, ANN etc) and what's the findings.

Response: Thank you for your valuable suggestions, we have made the changes in the manuscript as per the suggestions. We have added Table 2 with the details suggested by the reviewer.

Round 2

Reviewer 2 Report

The paper is really improved 

Author Response

Dear Reviewer,
Greetings of the day!

Thank you for your kind words of appreciation and accepting our revisions

Thank you and regards

Sincerely yours,

Dr. Farraj Albalawi (Corresponding Author)

Reviewer 2

The paper is really improved 

Response: Thank you for accepting our revisions
